# Tricuspid Regurgitation Velocity/Tricuspid Annular Plane Systolic Excursion (TRV/TAPSE) Ratio as a Novel Indicator of Disease Severity and Prognosis in Patients with Precapillary Pulmonary Hypertension

**DOI:** 10.3390/diseases11030117

**Published:** 2023-09-08

**Authors:** Weronika Topyła-Putowska, Michał Tomaszewski, Agnieszka Wojtkowska, Agnieszka Styczeń, Andrzej Wysokiński

**Affiliations:** Department of Cardiology, Medical University of Lublin, 20-059 Lublin, Poland

**Keywords:** echocardiography, pulmonary hypertension, tricuspid annular plane systolic excursion, tricuspid regurgitation velocity, prognosis

## Abstract

Background: Tricuspid annular plane systolic excursion (TAPSE) and tricuspid regurgitation velocity (TRV) are two echocardiographic parameters with prognostic value in patients with pulmonary hypertension (PH). When analyzed concurrently as the TRV/TAPSE ratio, they allow the ventricular–pulmonary artery coupling (RVPAC) to be assessed. This could better predict disease severity in patients with PH. Objective: Our study aimed to evaluate the prognostic value of the TRV/TAPSE ratio echocardiographic parameter in adults with precapillary PH. Methods: This study included 39 patients (74% women; average age, 63 years) with precapillary PH (pulmonary arterial hypertension and chronic thromboembolic PH) The mean follow-up period was 16.6 ± 13.3 months. Twelve patients (31%) died during the observation time. We measured TAPSE as a surrogate of RV contractility and TRV reflecting RV afterload, while ventricular–arterial coupling was evaluated by the ratio between these two parameters (TRV/TAPSE). To assess disease progression and the patient’s functional capacity, the World Health Organization functional class (WHO FC) was determined. Patient physical capacity was also evaluated using the 6 min walk test (6MWT). The analysis included values of N-terminal prohormone brain natriuretic peptide (NT-proBNP), which were taken routinely during the follow-up visit. Results: The mean calculated TRV/TAPSE ratio was 0.26 ± 0.08 m/s/mm. Upon comparison of the TRV/TAPSE ratio to the disease prognostic indicators, we observed a statistically significant correlation between TRV/TAPSE and the results of the WHO FC, 6MWT, and NT-proBNP. The TRV/TAPSE ratio is thus a good predictor of mortality in PH patients (AUC, 0.781). Patients with a TRV/TAPSE ratio > 0.30 m/s/mm had a shorter survival time, with log-rank test *p* < 0.0001. Additionally, ROC analysis revealed higher AUC for TRV/TAPSE than for TAPSE and TRV alone. Conclusions: TRV/TAPSE is a promising practicable echocardiographic parameter reflecting RVPAC. Moreover, TRV/TAPSE could be viable risk stratification parameter and could have prognostic value in patients with PH.

## 1. Introduction

Pulmonary hypertension (PH) is a disease characterized by elevated blood pressure in the pulmonary arteries. PH is not a homogeneous clinical or pathological entity. It most often occurs in the course of diseases of the left heart and lungs. Currently, there are five subtypes of PH, including pulmonary arterial hypertension (PAH), PH associated with left heart disease, PH associated with lung disease, chronic thromboembolic PH (CTEPH), and PH with unclear and/or multifactorial mechanisms [1]. PAH is distinguished from the other types by an aggressive course and poor prognosis but, paradoxically, also by the relatively best-known pathogenesis [2]. In PAH, all elements of the vascular wall are remodeled and accompanied by vasoconstriction and narrowing of their lumen as well as elements of inflammation and thrombosis [3,4]. This leads to a significant increase in pulmonary vascular resistance and, consequently, a significant pressure overload of the right ventricle, right ventricular heart failure, and death of the patient [5]. According to the new hemodynamic definition, PH is diagnosed when the mean pulmonary artery pressure (mPAP) in the hemodynamic examination is >20 mm Hg. The diagnosis of PAH is made when the pulmonary vascular resistance (PVR) is >2 WU and the pulmonary wedge pressure is ≤15 mmHg, which is precapillary PH. [6]. PAH is considered a rare disease, with an incidence estimated at 48–55 cases/million adults [7]. It includes idiopathic, inherited, and drug- or toxin-induced forms and is associated with certain diseases, including connective tissue diseases, congenital heart defects with systemic–pulmonary leakage, human immunodeficiency virus (HIV) infection, and portal hypertension [1,6].

CTEPH, like PAH, is classified as precapillary PH. It develops as a result of the obliteration of the lumen of the pulmonary vessels caused by emboli and/or thrombosis in situ. In the chronic form, thromboembolic masses undergo secondary fibrosis, which leads to reduced lumen and increased resistance in the pulmonary circulation [8]. CTEPH is a rare disease, although its incidence seems to be underestimated. The incidence in the general population is 3–30 cases/million adults [9,10].

Due to the crippling nature of the disease and the high mortality rate, it is important to make a timely diagnosis, implement effective treatment, and constantly monitor the patient’s condition. According to the latest guidelines, in order to monitor the clinical condition of the patient, a four-stage risk stratification scale was proposed. It is based on the World Health Organization functional class (WHO-FC), the 6 min walk test (6MWT), and B-type natriuretic peptide (BNP) or N-terminal (NT)-proBNP, and it categorizes patients as low, medium-low, medium-high, or high risk [6].

Echocardiography is one of the most frequently performed imaging modalities used to assess the structure and function of the heart. It is an extremely useful test to assess the probability of PH. It provides information about the possible etiology of PH and its severity and prognosis, allowing for the planning of observation and treatment. From a pathophysiological point of view, in patients with PH, the structure and systolic and diastolic functions of the right ventricle (RV), with a measurement of tricuspid regurgitation, the size of the right atrium and inferior vena cava, and the pulmonary artery pressure, should be assessed [11]. New echocardiographic parameters that could reliably predict prognosis in this group of patients are still being sought.

In recent studies, longitudinal systolic RV function indicators, such as TAPSE, and RV pressure afterload indicators, such as TRV, have been analyzed in combination. Also, a novel parameter, the TRV/TAPSE ratio, demonstrated a strong correlation with the New York Heart Association (NYHA) performance class in pediatric patients with PH [12].

This study aimed to assess the prognostic value of the TRV/TAPSE ratio echocardiographic parameter in adults with PH.

## 2. Methods

### 2.1. Population Characteristics

The study group included 39 adult patients (29 females and 10 males) with PAH or CTEPH treated at a single referral center. The inclusion criteria included a diagnosis of PAH or CTEPH after right heart catheterization based on the current guidelines of the European Society of Cardiology [6,13] and after the exclusion of left heart disease, lung diseases, and other forms of PH. The baseline demographic and clinical characteristics of the patients were also evaluated. Patients were enrolled in this study between November 2019 and June 2022, while the follow-up period ended on 31 December 2022, with a minimum observation period of six months attained. The Bioethics Committee of the Medical University of Lublin approved this study (number KE-0254/329/2019). Twelve deaths occurred during the follow-up. The cause of all deaths was an exacerbation of right heart failure in the course of PH.

To assess disease progression, the patient’s functional capacity was ascertained according to the World Health Organization WHO classification. Assessment of the amount of exertion that caused clinical symptoms (shortness of breath, fatigue, chest pain, or feeling faint) determined the functional capacity class. The WHO functional class (WHO FC) was decided by two independent physicians on the day of the echocardiographic examination.

Patient physical capacity was evaluated using the 6 min walk test (6MWT), which measured the distance the patient could cover in 6 min of steady walking, with simultaneous measurement of oxygen saturation and blood pressure. The 6MWT was performed after each echocardiographic examination on the same day.

Most measurements of N-terminal prohormone brain natriuretic peptide (NT-proBNP) were taken routinely during the follow-up visit on the day of the echocardiographic examination. In the absence of the result from the day of the echocardiography, NT-proBNP determined at the subsequent follow-up visit/hospitalization was used for the analysis. All NT-proBNP values used were marked +/− 40 days from the date of the echocardiographic examination.

### 2.2. Echocardiography

The gold standard for assessing the hemodynamic parameters of the pulmonary circulation is to measure the pressure in the right heart and the pulmonary vessels in cardiac catheterization. Although this test can diagnose PH, it is invasive and expensive. Therefore, among patients with suspected PH, transthoracic echocardiography (TTE) is used as the first-choice test for the noninvasive evaluation of pulmonary vascular hemodynamics and the initial assessment of RV, RA, and PA pressures [14].

Right atrial pressure (RAP) is usually estimated from the dimension and collapse ratio of the inferior vena cava (IVC) in the subcostal view [15]. Typically, the width of the IVC varies between 15 and 21 mm, and IVC collapse on inspiration should exceed 50%. Moderately elevated RA pressure exceeds 5 mmHg, and high pressure exceeds 10 mmHg [16]. It is considered that if there is no RV outflow tract stenosis, the right ventricular systolic pressure (RVSP) is equal to the pulmonary artery systolic pressure (PASP) [17]. In daily practice, the calculation of PASP is based on a simplified Bernoulli equation applied to peak tricuspid regurgitation velocity (TRV). One should aim to measure TRV in several projections, seeking the best image quality and maximum velocity in continuous Doppler and avoiding excessive enhancement and artifacts. According to the equation, PASP = 4 (TRV)^2^ +RAP [18]. Longitudinal myocardial fibers are mainly responsible for right ventricular contraction. Contraction of these fibers also causes movement of the tricuspid valve ring, which moves toward the apex of the ventricle in systole and toward the atrium in diastole. Therefore, the amplitude of the tricuspid valve annular systolic motion (TAPSE), measured in M-mode presentation from 4CH projection, reflects the RV systolic function [19]. Because the measurement of TAPSE is uncomplicated and little affected by image quality with a high prognostic value, it is recommended to be determined in all patients with PAH to assess RV systolic function [6].

Transthoracic echocardiography using a Philips EPIQ 7G (Epiq Elite Diagnostic Ultrasound System) with 2.5–3.5 MHz transducers was performed by one investigator (AW), following the European Association of Cardiovascular Imaging recommendations. Patients were examined in the left lateral position. TAPSE, measured as the difference between the displacement of the lateral tricuspid valve annulus from end-diastole to end-systole, was evaluated in the M-mode presentation and the standard apical 4-chamber view (4CV). TRV measurement used Continuous Wave Doppler focused on the regurgitation wave of the tricuspid valve in the 4CV view. The mean values of three cardiac cycles in sinus rhythm or five cycles in cases of atrial fibrillation were calculated. TRV/TAPSE ratio was calculated and expressed as m/s/mm units.

### 2.3. Statistical Analysis

Statistical analysis was performed using GraphPad Prism 9 (GraphPad Prism v9.0, San Diego, CA, USA). Data expressed on a quantitative scale were presented as mean with SD. Data expressed on a qualitative scale were presented as the number and percentage of the sample. Depending on the result of the Shapiro–Wilk test (assessment of compliance with the normal distribution), Pearson’s or Spearman correlation analysis was applied. Additionally, one-way ANOVA with Tukey’s post hoc test and the Kruskal–Wallis test with Dunn’s post hoc test were used. Furthermore, the receiver operating characteristic (ROC) curves were analyzed to obtain the cut-off values of tested variables that best distinguished patient survival, and the Kaplan–Meyer method was employed to plot survival curves. A log-rank test was also used to compare survival curves with the Cox proportional hazard test. Results were considered statistically significant when *p* < 0.05.

## 3. Results

### 3.1. General Results

The study group consisted of thirty-nine patients (twenty-nine females and ten males) with precapillary pulmonary hypertension, with a mean age of 63.1 ± 15.9 years (idiopathic pulmonary arterial hypertension, *n* = 12; connective tissue diseases PAH, *n* = 7; PAH associated with congenital defects, *n* = 13; portopulmonary hypertension, *n* = 1; chronic thromboembolic pulmonary hypertension, *n* = 6). Twelve patients (31%) died during the observation time, while the rest received PH-specific treatment during the study period. Eighteen patients were in the WHO functional class IV (46%), fifteen were in class III (38%), and six were in class II (15%).

The median NT-proBNP value was 2862 ± 3971 pg/mL, and the median 6MWT distance was 244 ± 163 m. The mean follow-up period was 16.6 ± 13.3 months.

All patients in this study were on a specific treatment for PAH and CTEPH. Most of the PAH patients were on a combined treatment. All CTEPH patients were treated with stimulators of soluble guanylate cyclase. The general characteristics of the study patients are shown in Table 1.

### 3.2. Echocardiographic Parameters

In the study group, the mean TAPSE was 17.4 ± 4 mm (female—17.124 ± 4.301 mm; male—18.200 ± 3.011 mm) and the mean TRV was 4.23 ± 0.72 m/s (female—4.184 ± 0.778 m/s; male—4.370 ± 0.505 m/s). The mean calculated TRV/TAPSE ratio was 0.26 ± 0.08 m/s/mm (female—0.263 ± 0.083 m/s/mm; male—0.249 ± 0.060 m/s/mm). Other echocardiographic parameters are presented in Table 2. The TRV/TAPSE ratio was compared to the disease prognostic indicators, the WHO FC and the 6MWT, as well as NT-proBNP values. Significantly statistically lower TRV/TAPSE values were found in the WHO II group compared to the WHO IV group (median 0.186 vs. 0.266; *p* = 0.0313). There were no statistically significant differences between the other groups (the power of the test was 0.618) (Figure 1).

In this study, we observed a significant negative correlation between TRV/TAPSE and the results of the 6MWT test (*r* = −0.37; *p* = 0.0193), as shown in Figure 2.

Furthermore, our study revealed that TRV/TAPSE correlated positively with the NT-proBNP values (*r* = 0.42; *p* = 0.0292) (Figure 3).

Our analysis indicated that there were no statistically significant differences in the TRV/TAPSE values between the groups of patients of different etiologies (IPAH, CHD, CTEPH, or CTDPAH) (the power of the test was 0.618) (Figure 4).

The receiver operating characteristic investigation disclosed that the TRV/TAPSE ratio is a good predictor of mortality in PH patients (AUC, 0.781). Additionally, the ROC analysis revealed higher AUC for TRV/TAPSE than for TAPSE and TRV alone. The ROC curve for the TAPSE variable has an AUC of 0.756 (*p* = 0.0115). The optimal cut-off point for predicting mortality was set at 16.50 mm (sensitivity 75.00%; specificity 70.37%). The ROC curve for the TRV variable has an AUC of 0.6049 (*p* = 0.3009), and one cut-off point was set at 4.43 m/s (sensitivity 66.67%; specificity 59.26%). The ROC curve for the TRV/TAPSE variable has an AUC of 0.781 (*p* = 0.0056). Two cut-off points were set, the first at 0.247 m/s/mm (sensitivity 75.00%; specificity 59.26%) and the second at 0.300 m/s/mm (sensitivity 66.67%; specificity 92.59%). Survival analyses were performed by taking both proposed values as the dividing point (Figure 5).

In the group of patients with TRV/TAPSE below 0.300 m/s/mm, no quartiles or medians of survival times were determined. In the group of patients with TRV/TAPSE equal to and above 0.300 m/s/mm, the lower quartile of survival time was 44 days, the median was 91 days, and the upper quartile was 187 days. The risk of death was found to be 23.880 times higher in the group with TRV/TAPSE equal to 0.300 m/s/mm and above (HR = 23.880; *p* < 0.0001) (the power of the test was 0.997) (Figure 6).

The schematic representation of the result is presented in Figure 7.

## 4. Discussion

Pulmonary hypertension (PH) is a condition where there is an abnormal increase in pressure in the pulmonary bed due to pulmonary vascular disease, lung parenchyma, and cardiac disease. Pulmonary arterial hypertension (PAH) is diagnosed in patients who have pre-capillary pulmonary hypertension on hemodynamic examination, defined as mPAP ≥ 20 mmHg, pulmonary artery wedge pressure (PAWP) ≤ 15 mm Hg, and pulmonary vascular resistance (PVR) > 3 Wood units (WU) [6]. PAH is one of the rare diseases, but it shows a steady upward trend. The natural course of the disease is severe, with a mean survival in idiopathic pulmonary hypertension of 2.8 years and survival of patients in WHO functional class IV not exceeding 6 months [20]. PAH poses a major diagnostic and therapeutic challenge due to progressive proliferative changes in the pulmonary arteries that lead to an increase in pulmonary resistance. Due to the nonspecific symptoms of the disease, the diagnosis is usually made quite late, when the structural changes of the pulmonary vasculature have already advanced. Without appropriate treatment, the disease leads to right heart failure and death. Although a number of medications slow or stop the course of PAH, it is still an incurable disease and, when pharmacotherapy is ineffective, this may be an indication for lung transplantation.

In assessing PH severity, several clinical, hemodynamic and biochemical parameters require determination. Echocardiography plays a crucial role in PH diagnosis, as it accurately assesses the correlation between RV contractility and RV afterload and the right ventricular–pulmonary artery coupling (RVPAC), while it is readily available, non-invasive and of low-cost [21]. It can also be applied to estimate disease severity in clinical practice, guide the selection of therapeutic strategies and monitor treatment effectiveness. Furthermore, it enables the assessment of RV overload and pressure in the RV and the pulmonary artery. As such, many echocardiographic parameters have proven prognostic value in PH [22].

TAPSE is one of the most recognized parameters reflecting RV systolic function [19], and it correlates with the WHO functional capacity class and NT-proBNP levels in PAH patients [23,24]. Furthermore, the TAPSE measurement technique is uncomplicated, has little dependence on image quality, and has a high prognostic value. Therefore, it is recommended for all PH patients. During the echocardiographic assessment of a PH patient, measuring systolic pulmonary artery pressure (sPAP) based on TRV has substantial prognostic value [25]. According to Bernoulli’s equation, sPAP = 4 (TRV)^2^ + right atrial pressure (RAP), using TRV as an indicator of pulmonary artery pressure is considered a prognostic factor of PH and plays a vital role in the current PH diagnostic algorithm [6,26].

Studies have shown, however, that over-reliance on one parameter for assessing the course of PH is an insufficient strategy [27]. Therefore, currently, RVPAC evaluation seeks to use several parameters simultaneously. Examples of new parameters recognized in the recent literature as particularly important in RVPAC assessment are TAPSE/sPAP [28,29] and TAPSE x pulmonary acceleration time (pACT) [30]. Referring to RVPAC idea, the TRV/TAPSE ratio reflects both the pressure in the pulmonary artery and the systolic function of the RV. In a single-center retrospective analysis including 102 patients with a diagnosis of PAH, using COMPERA and FPHN, it was shown that TAPSE/TRV and TAPSE/sPAP were the most powerful markers of prognosis. TAPSE/TRV or TAPSE/sPAP improved risk stratification in patients at intermediate risk [31].

This study aimed to determine the prognostic value of a novel alternative RVPAC assessment parameter, the TRV/TAPSE ratio, in adult PH patients. As such, the correlations between the TRV/TAPSE ratio and indicators of disease progression, including the WHO FC class, the 6MWT, NT-proBNP levels, and survival, were evaluated.

A version of the NYHA functional class modified for PH by the WHO monitors the main PH symptom—the deterioration of physical fitness. Studies have revealed that the WHO FC correlates with PH patient life expectancy. Herein, class I and II patients have a better prognosis, while classes III and IV are associated with a poorer prognosis [32]. Savarese et al. [33] conducted a meta-analysis of the results of 22 randomized clinical trials for assessing the association between 6MWT and the occurrence of a composite endpoint, which included death, hospitalization for PAH, and the need for lung or lung and heart transplantation. No correlation was found between the change in distance compared to the baseline study and the risk of the endpoint. However, a number of studies have indicated that improvements found in the 6MWT were paralleled by improvements in hemodynamic parameters assessed at the time of right heart catheterization [34]. The 6MWT is undoubtedly a very useful yet simple test to assess and compare exercise tolerance in patients undergoing targeted treatment for PAH. The disadvantages of this test are the lack of standardization and the influence of gender, height, age, and patient motivation on the result obtained. Koestenberger et al. noted that TRV/TAPSE strongly correlated with NYHA functional class in pediatric patients with PH [12]. Furthermore, in clinical practice, the 6MWT assesses physical fitness, and patients who walked > 400 m in the 6MWT had a better long-term prognosis [35]. According to our analysis, TRV/TAPSE values significantly increased with a deterioration in the WHO functional capacity class in adult PH patients. In addition, TRV/TAPSE strongly negatively correlated with the 6MWT. Therefore, TRV/TAPSE may be an indicator of disease severity.

Biochemical indicators of RV overload include BNP concentration and NT-proBNP levels. Moreover, the risk of PAH patient death increases with increased NT-proBNP plasma concentration [36,37]. A study of a group of 25 patients with pulmonary hypertension showed a negative correlation between the NT-proBNP levels and RV ejection fraction assessed by cardiac MRI [38]. Surveillance of natriuretic peptide levels in patients with PAH is very useful in determining the intensification of the treatment and thus can prevent an unfavorable course of the disease. This study demonstrated a strong correlation between TRV/TAPSE and NTpro-BNP levels. Accordingly, NT-proBNP values increased concurrently with increasing TRV/TAPSE values.

In the current study, the risk of death increased for patients with a higher TRV/TAPSE ratio. The Kaplan–Maier probability of survival analysis demonstrated that patients with a TRV/TAPSE ratio > 0.300 m/s/mm had a shorter survival time. Furthermore, receiver operating characteristic (ROC) curve analysis showed that the TRV/TAPSE ratio is a better predictor of PH-related death than TAPSE or TRV alone. Indeed, the area under the curve (AUC) was significantly higher for the TRV/TAPSE ratio than that for both TAPSE and TRV individually. Therefore, TRV/TAPSE may be a prognostic indicator in the course of PH in adult patients. The novel TRV/TAPSE ratio parameter correlated with disease severity indicators (including the WHO FC class, the 6MWT, NT-proBNP levels, and survival) in adults with PAH or CTEPH. Based on these results, the TRV/TAPSE ratio is a viable risk stratification parameter that can identify patients with poor prognoses. The statistical analysis highlighted TRV/TAPSE as a non-invasive indicator of RVPAC. As such, the findings may have important clinical significance for risk stratification and prognosis of PH patients. The TRV/TAPSE ratio is readily available for clinical use as it is measurable using the standard M-mode and Doppler echocardiography. Unlike the strong prognostic predictor TAPSE/sPAP, the TRV/TAPSE ratio does not require RAP estimation. Therefore, it could be used alternatively to evaluate RVPAC in cases with difficulties visualizing the inferior vena cava, such as with obese patients [39].

## 5. Limitations

A limitation of this study is the heterogeneity of patients in the etiology of precapillary PH and the relatively small size of the population with PAH. The TRV/TAPSE ratio parameter should be tested on a more substantial group of patients to confirm its clinical utility. In addition, the value of the parameter and its association with a worse prognosis should be validated in multicenter studies on a larger group of patients. What is more, the study group did not include WHO class I patients. Therefore, the clinical utility of the parameter in patients with mild PH remains unexplored. Data from right heart catheterization were not included in this study, due to the considerable time between the last RHC and the inclusion of patients in this study.

## 6. Conclusions

In conclusion, the TRV/TAPSE ratio is a promising practicable echocardiographic parameter reflecting RVPAC. It strongly correlates with prognostic markers of PH, such as WHO FC class, the 6MWT, NT-proBNP levels, and survival, in adults with PH. Therefore, the TRV/TAPSE ratio may be a viable risk stratification parameter, and it may have prognostic value in patients with PH.

## Figures and Tables

**Figure 1 diseases-11-00117-f001:**
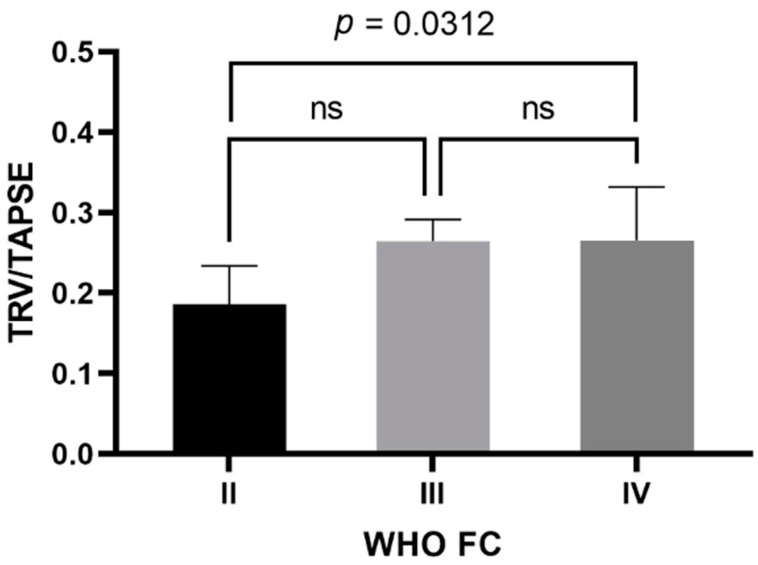
TRV/TAPSE level in groups of patients with WHO II, III, and IV class. One-way ANOVA with Tukey’s post hoc test. TRV/TAPSE ratio, tricuspid regurgitation velocity/tricuspid annular plane systolic excursion ratio; WHO FC—World Health Organization functional class; ns, not significant.

**Figure 2 diseases-11-00117-f002:**
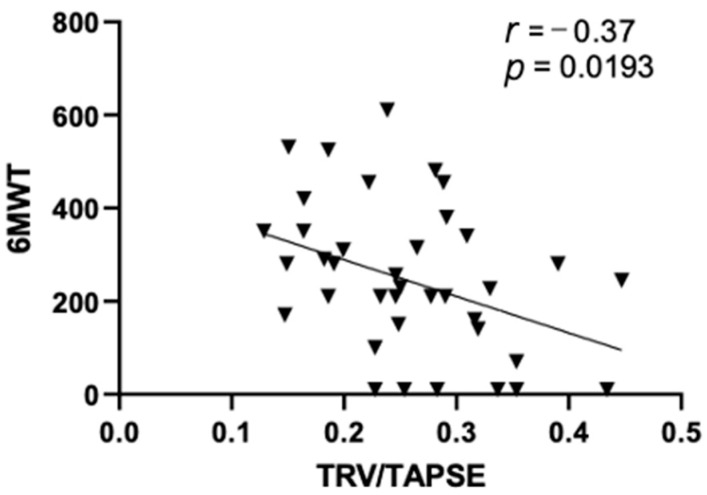
Pearson correlations between echo 6MWT and TRV/TAPSE ratio (*r* = −0.37; *p* = 0.0193). 6MWT, 6 min walk test; TRV/TAPSE ratio, tricuspid regurgitation velocity/tricuspid annular plane systolic excursion ratio.

**Figure 3 diseases-11-00117-f003:**
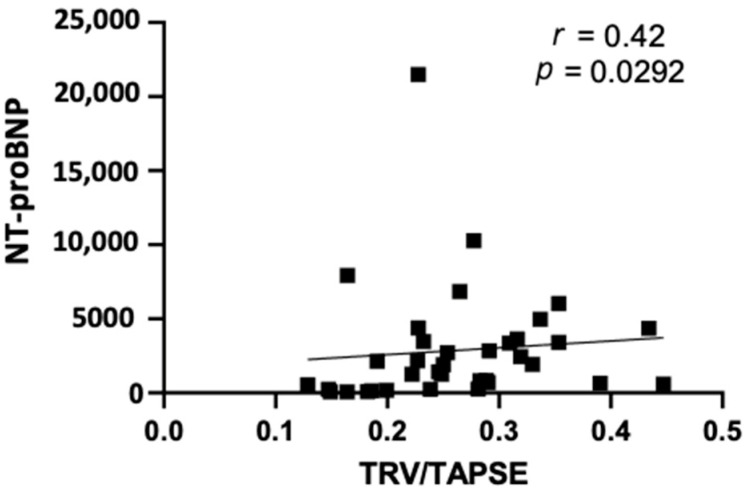
Spearman correlation between echo NT-proBNP and TRV/TAPSE ratio (*r* = 0.42; *p* = 0.0292). NT-proBNP, N-terminal pro- -B-type natriuretic peptide; TRV/TAPSE ratio, tricuspid regurgitation velocity/tricuspid annular plane systolic excursion ratio.

**Figure 4 diseases-11-00117-f004:**
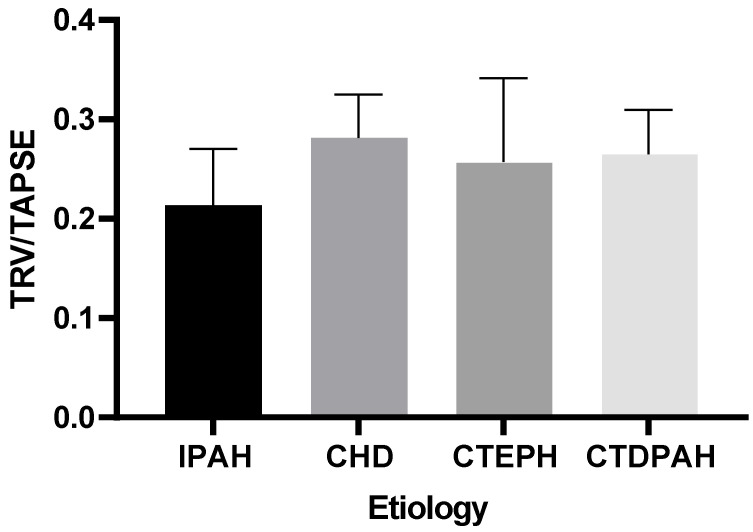
TRV/TAPSE levels in groups of different etiologies according to the Kruskal–Wallis test with Dunn’s post hoc test. TRV/TAPSE ratio, tricuspid regurgitation velocity/tricuspid annular plane systolic excursion ratio; IPAH, idiopathic pulmonary arterial hypertension; CHD, congenital heart disease pulmonary arterial hypertension; CTEPH, chronic thromboembolic pulmonary hypertension; CTDPAH, connective tissue disease-associated pulmonary arterial hypertension.

**Figure 5 diseases-11-00117-f005:**
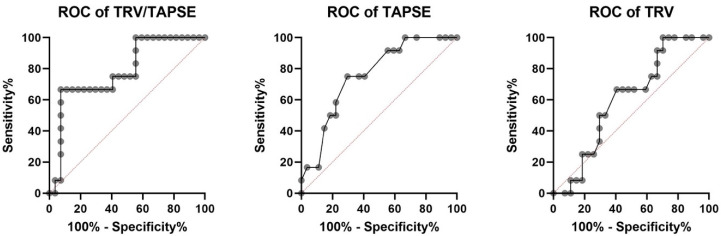
Receiver operating characteristic (ROC) analysis of TRV/TAPSE, TAPSE, and TRV for the endpoint of all-cause mortality in 39 patients with precapillary pulmonary hypertension (all deaths *n* = 12). AUC for TRV/TAPSE AUC of 0.756 (*p*= 0.0115). TAPSE, tricuspid annular plane systolic excursion; TRV, tricuspid regurgitation velocity; TRV/TAPSE ratio, tricuspid regurgitation velocity/tricuspid annular plane systolic excursion ratio.

**Figure 6 diseases-11-00117-f006:**
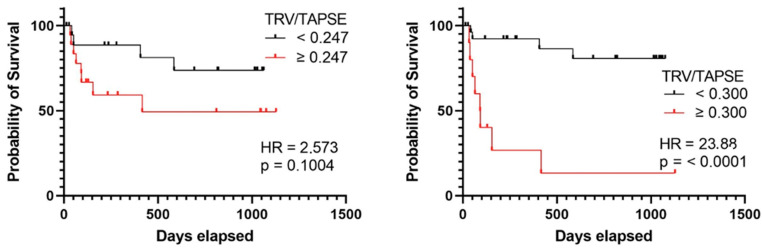
Kaplan–Meier curves presenting deterioration-free survival in precapillary PH patients based on TRV/TAPSE ratio ≥ or <0.247, log-rank test, *p* value = 0.1004, hazard ratio = 2.573 (the power of the test was 0.535), and TRV/TAPSE ratio ≥ or <0.300. Log-rank test, *p* value < 0.0001. Hazard ratio = 23.880 (the power of the test was 0.997). TRV/TAPSE ratio, tricuspid regurgitation velocity/tricuspid annular plane systolic excursion ratio.

**Figure 7 diseases-11-00117-f007:**
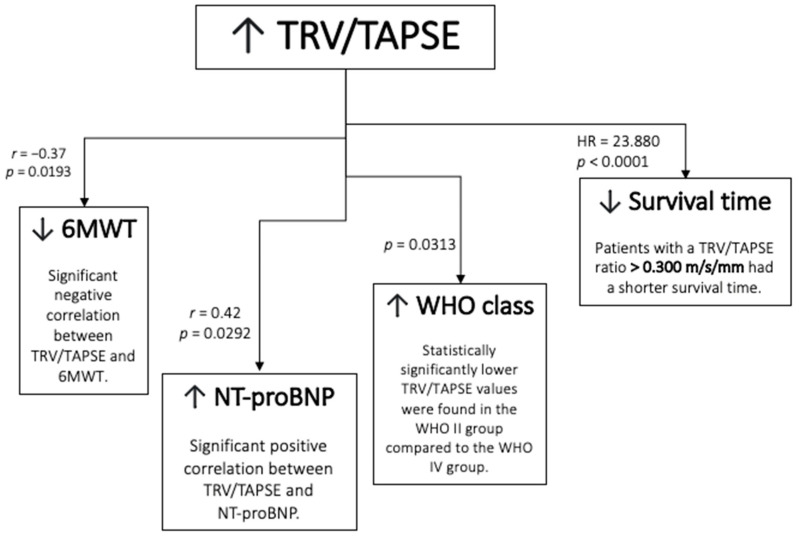
Schematic representation of the results. TRV/TAPSE ratio, tricuspid regurgitation velocity/tricuspid annular plane systolic excursion ratio; 6MWT, 6 min walk test; NT-proBNP, N-terminal pro- -B-type natriuretic peptide; WHO FC—World Health Organization functional class; HR, hazard ratio.

**Table 1 diseases-11-00117-t001:** Clinical characteristics of the study patients. WHO FC, World Health Organization functional class; 6MWT, 6 min walk test; NT-proBNP, N-terminal pro-B-type natriuretic peptide; IPAH, idiopathic pulmonary arterial hypertension; CTD-PAH, PAH associated with connective tissue disease; CHD-PAH, congenital heart disease PAH; PoPH, portopulmonary hypertension; CTEPH, chronic thrombo-embolic PH.

Clinical Characteristics of the Study Patients	
**General characteristic**	
Age, years	63.1 ± 15.9
Female gender, % (*n*)	74% (29)
BMI, kg/m2	23.5 ± 3.1
Resting heart rate, beats/min	83 ± 14
II WHO FC, % (*n*)	15.4% (6)
III WHO FC, % (*n*)	38.5% (15)
IV WHO FC, % (*n*)	46.2% (18)
6MWT distance, m	244 ± 163
NT-proBNP, pg/ml	2862 ± 3971
**PH etiology**	
IPAH, % (*n*)	30.8% (12)
CTD-PAH, % (*n*)	33.3% (13)
CHD-PAH, % (*n*)	17.9% (7)
PoPH, % (*n*)	2.6% (1)
CTEPH, % (*n*)	15.4% (6)
**Comorbidities**	
Hypertension, % (*n*)	66.7% (26)
Diabetes, % (*n*)	41.0% (16)
Obesity, % (*n*)	23.1% (9)
Hyperlipidemia, % (*n*)	56.4% (22)
Chronic kidney disease, % (*n*)	17.9% (7)
Heart failure, % (*n*)	41.0% (16)
Atrial fibrillation, % (*n*)	43.6% (17)
Ischemic heart disease, % (*n*)	28.2% (11)
**PH treatment**	
Endothelin receptor antagonist, % (*n*)	61.5% (24)
Phosphodiesterase-5 inhibitors, % (*n*)	82.1% (32)
Prostanoids, % (*n*)	43.6% (17)
Stimulator of soluble guanylate cyclase, % (*n*)	7.7% (3)
Agonists of the prostacyclin receptor, % (*n*)	15.4% (6)
Diuretics, % (*n*)	43.6% (17)

**Table 2 diseases-11-00117-t002:** Echocardiographic parameters of the study group. LV EF, left ventricular ejection fraction; FAC, fractional area change; pAcT, pulmonary artery acceleration time; mPAP, mean pulmonary arterial pressure; TAPSE, tricuspid annular plane systolic excursion; RVSP, right ventricular systolic pressure; TRV, tricuspid regurgitation velocity; TRV/TAPSE, tricuspid regurgitation velocity/tricuspid annular plane systolic excursion ratio.

Echocardiographic Parameters	
LV EF, %	59.68 ± 6.52
FAC, %	36.7 ± 14.7
pAcT, m/s	82.06 ± 19.47
mPAP, mmHg	41.99 ± 8.59
TAPSE, mm	17.4 ± 4.0
RVSP, mmHg	79.12 ± 24.94
TRV m/s	4.23 ± 0.72
TRV/TAPSE, m/s:mm	0.26 ± 0.08

## Data Availability

Data are available on request.

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
