# Peer review of "Tricuspid Regurgitation Velocity/Tricuspid Annular Plane Systolic Excursion (TRV/TAPSE) Ratio as a Novel Indicator of Disease Severity and Prognosis in Patients with Precapillary Pulmonary Hypertension"

_diseases, 2023, doi:10.3390/diseases11030117_

Round 1

Reviewer 1 Report

The authors have aimed to determine the prognostic value of a novel alternative RVPAC assessment parameter, the TRV/TAPSE ratio, in adult PH patients. As such, the correlations between the TRV/TAPSE ratio and indicators of disease progression, including WHO FC class, the 6MWT, NT-proBNP levels and survival, were evaluated. The study showed that the risk of death increased for patients with a higher TRV/TAPSE ratio and patients with a TRV/TAPSE ratio > 0.300 m/s/mm had a shorter survival time.

I have a few technical comments:

What was the criteria for selecting the sample size? Why the gender distribution among the population selected was different?

In the inclusion criteria different kinds of comorbidities are also present. Is there any relation with those diseases with the manifestation of PH or death of patients?

There ae some grammatical errors in the manuscript which needs to be addressed.

The authors can include some high- quality schematic representation of the results obtained since Diseases is a flagship journal of MDPI.

I recommend minor revision.

Grammatical errors are present and needs to be addressed.

Author Response

Dear Reviewer,
we would like to thank for the comments. We attach the revision of our manuscript. We hope that Editor and Reviewers will find our manuscript suitable for publication in Diseases journal as it provides a novel, clinically useful parameter for evaluating patients with pre-capillary pulmonary hypertension

Point 1:
What was the criteria for selecting the sample size? Why the gender distribution among the population selected was different?
Response 1:
Dear Reviewer, 
     Pulmonary arterial hypertension (PAH) and chronic thromboembolic pulmonary hypertension (CTEPH) are considered rare diseases. According to Polish Registry of Pulmonary Hypertension, the estimated prevalence and annual incidence of PAH were 30.8/mln adults and 5.2/mln adults [1]. The period prevalence of CTEPH was 16.4 per million adults and the incidence rate was estimated at 3.96 new CTEPH patients per million per year [2]. In our center, the Department of Cardiology, Medical University of Lublin, Poland is treated about 30 patients with PAH and CTEPH a year. Patients were enrolled in the study between November 2019 and June 2022, while the follow-up period ended on December 31st, 2022, with a minimum observation period of six months attained. During this period, we examined 39 patients who complied with the study's inclusion criteria.
     There is a general consensus that female gender is a risk factor for PAH. Registry data worldwide indicate an overall female predominance for PAH of 2–4 over men, depending on the PAH etiology [3], [4]. Also, in Polish registers female-to-male ratio of 2.3 in the total group of PAH patients and 2.5 in the group of patients with idiopathic PAH [1].  In case of CTEPH, the European registry shows the ratio of women vs men was 1:1, which is similar to what reported by the authors of Polish registry [2], [5], [6]. In the patient population of our center, the female-to-male ratio for both, PAH and CTEPH, is about 2.5-3:1, which reflects the mentioned registries.

Point 2:
In the inclusion criteria different kinds of comorbidities are also present. Is there any relation with those diseases with the manifestation of PH or death of patients?
Response 2:
Dear Reviewer, 
       in pulmonary hypertension (PH) patients, approximately three-quarters of patients have at least one comorbidity, with a higher number of comorbidities in patients 65 years of age and older [7]. Current data suggest that the presence of comorbidities in PH patients negatively affects treatment outcomes. Comorbidities can mask the manifestation of PH, resulting in diagnosis delays, as well as difficulty in assessing disease progression and treatment outcomes. Considering the multifactorial pathophysiology of PH, the presence of comorbidities can lead to difficulties in distinguishing group 1 PH (PAH) from other PH groups. Many comorbidities also contribute to the PH progression by increasing pulmonary artery pressure and cardiac output, so treatment of a comorbidity may also help reduce the PH severity [8].
     For instance, the most common comorbidity in patients with PH- systemic hypertension, increases the risk of cardiovascular morbidity and mortality, including coronary events and stroke [9]. Another example is obesity, the prevalence of which in patients with PAH is about 35%. It has been observed that there is increased mortality in patients with a BMI ≥40 kg/m2 with PAH under 65 years of age, and higher BMI (≥30 kg/m2) is associated with poorer WHO functional class [10].  Furthermore, it has been shown, that patients with PAH and diabetes have a significantly lower 10-year survival rate compared to patients without diabetes [11].  In conclusion, comorbidities can mask the PH symptoms and can lead to delayed diagnosis, disease progression and reduced survival.

Point 3:
There ae some grammatical errors in the manuscript which needs to be addressed.     

Response 3:
Following the Reviewer’s comment, we corrected grammatical errors found in the manuscript.

Point 4:
The authors can include some high- quality schematic representation of the results obtained since Diseases is a flagship journal of MDPI.

Response 4:
As proposed by the Reviewer, we added schematic representation of the results.

References:

1 Kopeć, G.; Kurzyna, M.; Mroczek, E., Chrzanowski, Ł., Mularek-Kubzdela, T., Skoczylas, I., Kuśmierczyk, B., Pruszczyk, P., Błaszczak, P., Lewicka, et al. Characterization of Patients with Pulmonary Arterial Hypertension: Data from the Polish Registry of Pulmonary Hypertension (BNP-PL). J Clin Med. 2020, 9, 173. doi:10.3390/jcm9010173

2. Kopeć, G.; Dzikowska-Diduch, O.; Mroczek E, Mularek-Kubzdela, T., Chrzanowski, Ł., Skoczylas, I., Tomaszewski, M., Peregud-Pogorzelska, M., Karasek, D., Lewicka, E., et al. Characteristics and outcomes of patients with chronic thromboembolic pulmonary hypertension in the era of modern therapeutic approaches: data from the Polish multicenter registry (BNP-PL). Ther Adv Chronic Dis. 2021, 12:20406223211002961. doi:10.1177/20406223211002961

3. Mair, K.M.; Johansen, A.K.; Wright, A.F.; Wallace, E.; MacLean, M.R. Pulmonary arterial hypertension: basis of sex differences in incidence and treatment response. Br J Pharmacol. 2014, 171, 567-79. doi: 10.1111/bph.12281. PMID: 23802760; PMCID: PMC3969073.

4. Batton, K.A.; Austin, C.O.; Bruno, K.A.; Burger, C.D.; Shapiro, B.P.; Fairweather, D. Sex differences in pulmonary arterial hypertension: role of infection and autoimmunity in the pathogenesis of disease. Biol Sex Differ. 2018, 18, 15. doi: 10.1186/s13293-018-0176-8.

5. Barco ,S.; Klok, F.A.; Konstantinides, S.V.; Dartevelle, P.; Fadel, E.; Jenkins, D.; Kim, N.H.; Madani, M.; Matsubara, H.; Mayer, E.; et al. Sex-specific differences in chronic thromboembolic pulmonary hypertension. Results from the European CTEPH registry. J Thromb Haemost. 2020, 18, 151-161. doi: 10.1111/jth.14629.

6. Kerr, K.M.; Auger, W.R.; Benza, R.L, et al. Preliminary data from the United States CTEPH Registry. J Heart Lung Transplant. 2017, 36, 19–20. https://doi.org/10.1016/j.healun.2017.01.039

7. Hjalmarsson, C.; Radegran, G.; Kylhammar, D.; Rundqvist, B.; Multing, J.; Nisell, M.D.; Kjellstrom, B. Impact of age and comorbidity on risk stratification in idiopathic pulmonary arterial hypertension. Eur Respir J. 2018, 51:1702310.

8. Lang, I.M.; Palazzini, M. The burden of comorbidities in pulmonary arterial hypertension. Eur Heart J Suppl. 2019, 21, 21-28. doi: 10.1093/eurheartj/suz205.

9. Aronow, W.S. Treatment of systemic hypertension. Am J Cardiovasc Dis 2012, 2, 160–170.

10. Weatherald, J.; Huertas, A.; Boucly, A.; Guignabert, C.; Taniguchi, Y.; Adir, Y.; Jevnikar, M.; Savale, L.; Jais, X.; Peng, M.; et al. Association between BMI and obesity with survival in pulmonary arterial hypertension. Chest. 2018, 154, 872–881

11. Benson, L.; Brittain, E.L.; Pugh, M.E.; Austin, E.D.; Fox, K.; Wheeler, L.; Robbins, I.M.; Hemnes, A.R. Impact of diabetes on survival and right ventricular compensation in pulmonary arterial hypertension. Pulm Circ. 2014, 4, 311–318.

Reviewer 2 Report

The authors examined 39 patients using echocardiography, specifically focusing on TRV/TAPSE ratio. They  observed statistically significant correlation between TRV/TAPSE and the results of WHO FC, 6MWT and NT-  proBNP. TRV/TAPSE ratio is also a good predictor of mortality in PH patients (AUC, 28 0.781). Patients with a TRV/TAPSE ratio > 0.30 m/s/mm had a shorter survival time with log-rank test p< 0.0001. They conclude that TRV/TAPSE is a promising practicable echocardiographic parameter reflect- ing RVPAC. Moreover, TRV/TAPSE could be viable risk stratification parameter and could have prognostic value in patients with PH. Scientifically and clinically this manuscript is very important, and can be publishable in this journal after they can well address the following major and minor concerns:

1. the study cohort size is very limited. They did discuss this limitation, but they have not performed power analysis. It is required to do it.

2. They have not included healthy controls, which is a big concern for this manuscript. I suggest them to include age and gender matched healthy controls.

3. They are required to show RVSP or mPAP values if possible.

4. Female and male PH patients may have difference in RV echo. I suggest them to do this comparisons.

5. When they performed echo, I believe that they should perform parasternal short or long-axis echo imaging, from which they can calculate LV systolic function data including EF, FS, CO. They may be able to measure RV fraction area change, RV free wall thickness, PAT or PAT/PET. If they have this data, they should present this data as well for further confirmation.

6. heart rates are very important for us to perform meaningful echocardiography. It seems that they have not provided this data.

7. two minor concerns: 16,6+/-13,3 should be 16.6 +/- 13.3; some reference format is not consistent.

good enough

Author Response

Dear Reviewer,

we would like to thank for the comments. We attach the revision of our manuscript. We hope that Editor and Reviewers will find our manuscript suitable for publication in Diseases journal as it provides a novel, clinically useful parameter for evaluating patients with pre-capillary pulmonary hypertension

Point 1:          
the study cohort size is very limited. They did discuss this limitation, but they have not performed power analysis. It is required to do it.  

Response 1:
As suggested by the Reviewer, we performed power analysis and added the results to the paper.

Point 2:
They have not included healthy controls, which is a big concern for this manuscript. I suggest them to include age and gender matched healthy controls.   

Response 2:
Dear Reviewer,

     In our study, a control group consisting of 23 healthy adults matched by age and gender to the study group, was also examined. In the control group, a clinical history was taken and a complete echocardiogram and 6- minute walk test (6MWT) were performed. However, NT-proBNP was not determined. This parameter was collected as part of routine hospital visits in the study group.
     In our manuscript, we focused on the TRV/TAPSE parameter, which we showed correlates with disease course and survival in patients with pre-capillary pulmonary hypertension. Since about half of the control subjects had no tricuspid regurgitation detected by echocardiography, the number of TRV measurements obtained was unrepresentative. In view of this, we waived statistical analysis of TRV/TAPSE in the control group.
      We agree with the reviewer's comment, that the presence of the control group in the statistical analysis would have significantly improved the paper's value.  In future studies, we plan to increase the size of the study and control group and better evaluate TRV/TAPSE in a population of healthy adults.

Point 3:
They are required to show RVSP or mPAP values if possible.            

Response 3:
As proposed by the Reviewer, we added RVSP and mPAP values in Table 2 of our manuscript.

Point 4:
Female and male PH patients may have difference in RV echo. I suggest them to do this comparisons.  

Response 4:   
Following the Reviewer’s suggestion, we have added TRV, TAPSE and TRV/TAPSE values for men and women.

Point 5:
When they performed echo, I believe that they should perform parasternal short or long-axis echo imaging, from which they can calculate LV systolic function data including EF, FS, CO. They may be able to measure RV fraction area change, RV free wall thickness, PAT or PAT/PET. If they have this data, they should present this data as well for further confirmation. 

Response 5:
Dear Reviewer,

In the echocardiograms we performed, several parameters were evaluated, mainly focusing on right ventricular function. Following the Reviewer’s suggestion, we added the values of RV fraction area change and PAT to the Table 2 of the manuscript.
Regarding left ventricular function, we assessed LV EF, which was added to the manuscript, as proposed by the Reviewer. 
Unfortunately, parameters such as, FS and CO and RV free wall thickness were not assessed in our study.  As suggested by the Reviewer, we will also include the mentioned parameters in future studies.

Point 6:

heart rates are very important for us to perform meaningful echocardiography. It seems that they have not provided this data.  

Response 6:
All patients had their heart rhythm evaluated three times on the day of the echocardiogram, during resting ECG, 6MWT and echocardiography.  As proposed by the Reviewer, we added the mean value of resting heart rate to the Table 1 of the manuscript.

Point 7:
two minor concerns: 16,6+/-13,3 should be 16.6 +/- 13.3; some reference format is not consistent. 

Response 7:
Following the Reviewer’s suggestion, we corrected grammatical errors found in the manuscript.

Round 2

Reviewer 2 Report

the authors have well addressed all of my questions